# Myosin 1C isoform A is a novel candidate diagnostic marker for prostate cancer

Aleena A. Saidova[1]*, Daria M. Potashnikova[1], Anna V. Tvorogova[2], Oxana V. Paklina[3], Evgeniy I. Veliev[4], Grigoriy V. Knyshinsky[3], Galiya R. Setdikova[3], Daniil L. Rotin[3], Ivan V. Maly[5], Wilma A. Hofmann[5], Ivan A. Vorobjev[1,2,6]

**1** School of Biology, Cell Biology and Histology Department, M.V. Lomonosov Moscow State University, Moscow, Russia, **2** A.N. Belozersky Institute of Physico-Chemical Biology, M.V. Lomonosov Moscow State University, Moscow, Russia, **3** Pathoanatomy Department, S.P. Botkin Clinical Hospital, Moscow, Russia, **4** Urology Department, S.P. Botkin Clinical Hospital, Moscow, Russia, **5** Department of Physiology and Biophysics, Jacobs School of Medicine and Biomedical Sciences, University at Buffalo, Buffalo, NY, United States of America, **6** Department of Biology, School of Sciences and Humanities, Nazarbayev University, Nur-Sultan, Kazakhstan

☯ These authors contributed equally to this work.

\* saidova@mail.bio.msu.ru

**Data Availability Statement:** All relevant data are within the paper and its Supporting Information files.

## Abstract

Early diagnosis of prostate cancer is a challenging issue due to the lack of specific markers. Therefore, a sensitive diagnostic marker that is expressed or upregulated exclusively in prostate cancer cells would facilitate diagnostic procedures and ensure a better outcome. We evaluated the expression of myosin 1C isoform A in 5 prostate cell lines, 41 prostate cancer cases, and 11 benign hyperplasias. We analyzed the expression of 12 surface molecules on prostate cancer cells by flow cytometry and analyzed whether high or low myosin 1C isoform A expression could be attributed to a distinct phenotype of prostate cancer cells. Median myosin 1C isoform A expression in prostate cancer samples and cancer cell lines was 2 orders of magnitude higher than in benign prostate hyperplasia. Based on isoform A expression, we could also distinguish clinical stage 2 from clinical stage 3. Among cell lines, PC-3 cells with the highest myosin 1C isoform A level had diminished numbers of CD10/CD13-positive cells and increased numbers of CD29 (integrin β1), CD38, CD54 (ICAM1) positive cells. The surface phenotype of clinical samples was similar to prostate cancer cell lines with high isoform A expression and could be described as CD10-/CD13- with heterogeneous expression of other markers. Both for cell lines and cancer specimens we observed the strong correlation of high myosin 1C isoform A mRNA expression and elevated levels of CD29 and CD54, suggesting a more adhesive phenotype for cells with high isoform A expression. Compared to normal tissue, prostate cancer samples had also reduced numbers of CD24- and CD38-positive cells. Our data suggest that a high level of myosin 1C isoform A is a specific marker both for prostate cancer cells and prostate cancer cell lines. High expression of isoform A is associated with less activated (CD24/CD38 low) and more adhesive (CD29/CD54 high) surface phenotype compared to benign prostate tissue.

**Funding:** This work was supported in part by the NCI of the National Institutes of Health under award number R21CA220155 to Wilma A. Hofmann and RFBR grant #17-5433009 awarded to Ivan Vorobjev. The funders had no role in study design, data collection and analysis, decision to publish, or preparation of the manuscript.

**Competing interests:** The authors have declared that no competing interests exist.

# Introduction

Early diagnosis of prostate cancer (PCa) is critically important due to its high incidence and related mortality [1]. A final diagnosis of prostate cancer in routine practice is based on the histological evaluation of transrectal ultrasound (TRUS) biopsies or surgical material. Diagnostic accuracy of this method remains low, as 30% of prostate cancers are missed, and many men without cancer undergo unnecessary invasive procedures [2]. Both MP-MRI [3] and increasing the number of core biopsies from 12 to 24 [4] were proposed to improve the sensitivity of the diagnostic procedures. However, the former approach is not always available in routine clinical practice, and the latter is associated with risk- and can cause life-threatening sepsis [5]. Heterogeneity of clinical symptoms highlights the need for a new generation of specific and sensitive PCa markers, which would decrease the invasiveness of diagnostic interventions and facilitate early diagnosis [6]. Among novel promising markers [7] isoform A of myosin 1C [8] has been described as a specific marker of prostate cell lines and mouse prostate cancer tissues [9]. Recent findings highlight the role of this protein in mechanoregulation of epithelial cell-cell adhesion [10] and stimulation of exosome secretion in prostate cancer cells [11]. In this study, we evaluated myosin 1C isoform A mRNA and protein expression levels in clinical samples of prostate cancer and benign prostate hyperplasia (BPH).

# Materials and methods

## Clinical samples

Forty-one patient with prostate cancer and eleven patients with BPH who were diagnosed in the Urology department of S. P. Botkin City Clinical Hospital were enrolled in this study from February 2019 to August 2020. Patient age was from 42 to 74 years (mean 57.9±8.5 years). Levels of prostate specific antigen (PSA) for PCa samples ranged between 0.80–44 ng/ml (mean 12.9 ±10.8 ng/ml); levels of PSA for BPH samples ranged from 1.8 to 10 ng/ml (mean 4.3±2.8 ng/ml). Gleason score was 6 for 12 patients, 7 for 22 patients, five patients had a Gleason score of 8, and two had a Gleason score of 9. Clinical data for all patients are summarized in S1 Table.

Samples for RTqPCR and flow cytometry were processed immediately after surgery. Cryostat sections were prepared from different parts of the post-surgical material; each section was evaluated by an experienced pathologist. All human material was obtained with written informed consent. All procedures were performed following the ethical standards of the responsible local committee (study approved at the MSU Biology Department Bioethical Committee protocol #127-d) and the Helsinki Declaration of 1975 as revised in 2008. Patients with previous prostate surgery or chemotherapy were excluded from the study. Staging of clinical samples was performed according to American Joint Committee on Cancer (8th edition) [12]. Examples of histological patterns for samples with different Gleason scores are provided in S1 Fig.

Formalin-fixed, paraffin-embedded (FFPE) samples of normal and cancerous human prostate were purchased from US Biomax, Inc. (Derwood, Maryland). These samples were collected under HIPPA-approved protocols with the donors' informed consent. Eosin- and hematoxylin-stained FFPE sections were imaged on a Leica DMRE microscope (Leica, Wetzlar) equipped with an RT Slider CCD camera (SPOT Imaging, Sterling Heights MI).

## Cell lines

Non-malignant prostate epithelial cell line (RWPE-1) (CRL116-09) and prostate cancer cell lines LNCaP (CRL-1740), DU145 (CRL-3356), 22Rv1 (CRL-2505), PC-3 (CRL-1435) were

obtained from American Type Culture Collection (ATCC, Manassas, VA, USA). RWPE-1 cells were cultured in serum-free keratinocyte medium with L-glutamine (Gibco-Thermo Fisher Scientific, Wlatham, MA, USA), supplemented with 1% gentamycin (Paneco, Russia), 0.05 mg/ml bovine pituitary extract (BPE) and 5 ng/ml human recombinant epidermal growth factor (EGF) (both from Gibco-Thermo Fisher Scientific, Waltham, MA, USA). Cancer cell lines were cultured in DMEM/F12 supplemented with 10% FBS, with 1% gentamycin and L-glutamine (Paneco, Moscow, Russia). Cells were grown in a humidified atmosphere with 5% $CO_2$ at 37˚C. Cells were routinely tested for mycoplasma, and only cells that were mycoplasma-free were used in the experiments.

## RNA extraction and cDNA synthesis

RNA was extracted from thawed suspensions of clinical samples and cell cultures using the RNeasy Mini Kit (Qiagen, Germantown, MD, USA) according to the manufacturer's instructions. The RNA concentration was measured using a NanoPhotometer (Implen, Munich, Germany), and its purity was assessed according to the A260/A280 and A260/A230 ratios. cDNA was transcribed using the ImProm-II AMV-Reverse Transcription Kit (Promega, Madison, WI, USA) according to the manufacturer's instructions. RNA was extracted from the FFPE samples (same tissue cores as used for histology) using the Absolutely RNA kit (Agilent, La Jolla, CA) according to the manufacturer's instructions and reverse-transcribed with Superscript III (Thermo Fisher Scientific, Waltham, MA, USA).

## Primers and real-time PCR

Real-time qPCR experiments were performed using CFX96 Touch cycler (Bio-Rad, Hercules, CA, USA). All samples were processed in triplicate. One sample of cDNA put into each PCR run served as an inter-run calibrator for combining data into one experiment. Primer details were reported previously [8,13]. Primer sequences are provided in S2 Table. Primers for isoform A specifically recognize human myosin 1C isoform A (NCBI reference sequence NM_001080779.1). Primers were purchased from Synthol (Moscow, Russia). Primer specificity was confirmed by melting curve analysis and detection of products with predicted length by electrophoresis on a 1.5% agarose gel. The reaction protocol included denaturation (95˚C, 10 min), followed by 40 amplification cycles (95˚C, 15 sec; 60˚C, 30 sec; 72˚C, 60 sec). The $C_t$ values were determined for real-time PCR curves by setting the threshold at 5 SD for each run. qPCR data were normalized according to Vandesompele et al. [14].

## Western blotting

Cells were lysed in RIPA buffer for 30 minutes on ice. Protein concentration was measured using Bradford reagent (Sigma-Aldrich, St. Louis, MO, USA). 50 μg of protein lysates were resolved in 12% SDS-PAAG gel and transferred to 0.2 μm nitrocellulose membrane (GE Healthcare, Chicago, IL, USA), blocked with 5% nonfat dry milk, 1X TBS, 0.1% Tween-20 for 40 minutes at RT. Membranes were incubated with primary antibodies against myosin 1C isoform A (custom antibody [8]) and alpha-tubulin (Cell Signaling Technology, Danvers, MA, USA; clone DM1A 2128, clone 9f3, lot7) at +4˚ overnight in 1% BSA in 1X TBS, 0.1% Tween-20. Mouse monoclonal antibody against N-terminal peptide of myosin 1C isoform A was described in Ihnatovych et al [8,9,15]. After washing, membranes were incubated with mouse or rabbit secondary antibodies linked with horseradish peroxidase (Cell Signaling Technology, Danvers, MA, USA) for 1 hour at RT. Proteins were visualized using Image Quant LAS 4000 system (GE Healthcare, Chicago, IL, USA).

## Flow cytometry

37 clinical samples (29 tumors and 8 benign hyperplasias) and 5 prostate cell lines were analyzed in suspension by multicolor flow cytometry using a FACSAria SORP instrument (BD Biosciences, San Jose, CA, USA).

Single-cell suspensions of solid prostate tissues were prepared using a MediMachine (BD Biosciences, San Jose, CA, USA) and stained for surface antigens with monoclonal antibodies, as presented in **S3 Table**. Unspecific binding was blocked by normal mouse serum (Abcam, Cambridge, MA, USA), cells were incubated with antibodies at RT for 20 minutes, washed once in PBS and analyzed. FACSDiva software (BD Biosciences, San Jose, CA, USA) was employed for data acquisition and analysis. The spectral compensation matrix was calculated automatically using single-stained controls. FACS profiles were obtained for each sample on the day of retrieval, typically in 4 hours after the extraction.

## Data analysis

Statistical analysis was performed in GraphPad Prizm 6 (GraphPad Software, San Diego, CA, USA) and R (R Core Team, 2019) with basic package "stats" ver. 3.6.0 (R Core Team, 2019), additional package "PMCMR" ver. 4.3 and graphic package "ggplot2" ver. 3.3.1. To analyze the results, Kruskal—Wallis analysis of variance was performed to evaluate the differences in the level of myosin 1C isoform A or CD-positive cell percentages at different cancer stages. For pairwise comparisons, we performed the Conover-Iman test from the "PMCMR" package with a 0.95 confidence level. To measure the relationship between the isoform A level and CD-positive percentage, we calculated Spearman correlation coefficients for each cancer stage. Images were finalized in AdobePhotoshop Software (Adobe Inc., San Jose, CA, USA)

# Results

## Myosin 1C isoform A is overexpressed in clinical prostate cancer samples

We first addressed the difference in myosin 1C isoform A expression between benign prostate hyperplasia and prostate cancer clinical samples. To describe the expression of isoform A in prostate cancers, we divided our dataset into 3 groups according to the clinical stage. The first group included 11 samples with benign prostate hyperplasia, which we formally denote as clinical stage 0 in our comparisons. The second group included 28 samples with clinical stage 2, and the third group consisted of 13 samples with clinical stage 3. In the combined dataset, the median of relative normalized isoform A mRNA expression in prostate cancer samples was 2.078 (range 0.102–14.89) compared to 0.062 in benign prostate hyperplasia samples (range 0.0003–0.386) (Fig 1A, Table 1). Thus, mRNA expression of myosin 1C isoform A in prostate cancer cells is 2 orders of magnitude higher than for benign prostate hyperplasia, and the differences are statistically significant ($p < 0.0001$, Kruskal-Wallis test). The difference in isoform A expression was statistically significant between benign hyperplasia (stage 0) and all clinical stages (Fig 1B, Kruskal-Wallis test, $p < 0.0001$). Notably, based on isoform A expression we could also distinguish stage 2 and stage 3 ($p = 0.004$, Mann-Whitney test). mRNA expression of isoform A in prostate cancer cell lines was similar to human prostate cancer tissues (median 2.167, range 0.359–4.174), and the isoform A mRNA expression in the RWPE-1 cell line, which is assumed to be a normal prostate tissue analog, is similar to that in BPH. Finally, we confirmed the differences in isoform A mRNA expression between prostate cancer and benign hyperplasia at the protein level (Fig 1C).

To test whether the uncovered pattern of expression could also be discerned in archival prostate cancer tissue samples, we performed the analysis on formalin-fixed, paraffin-

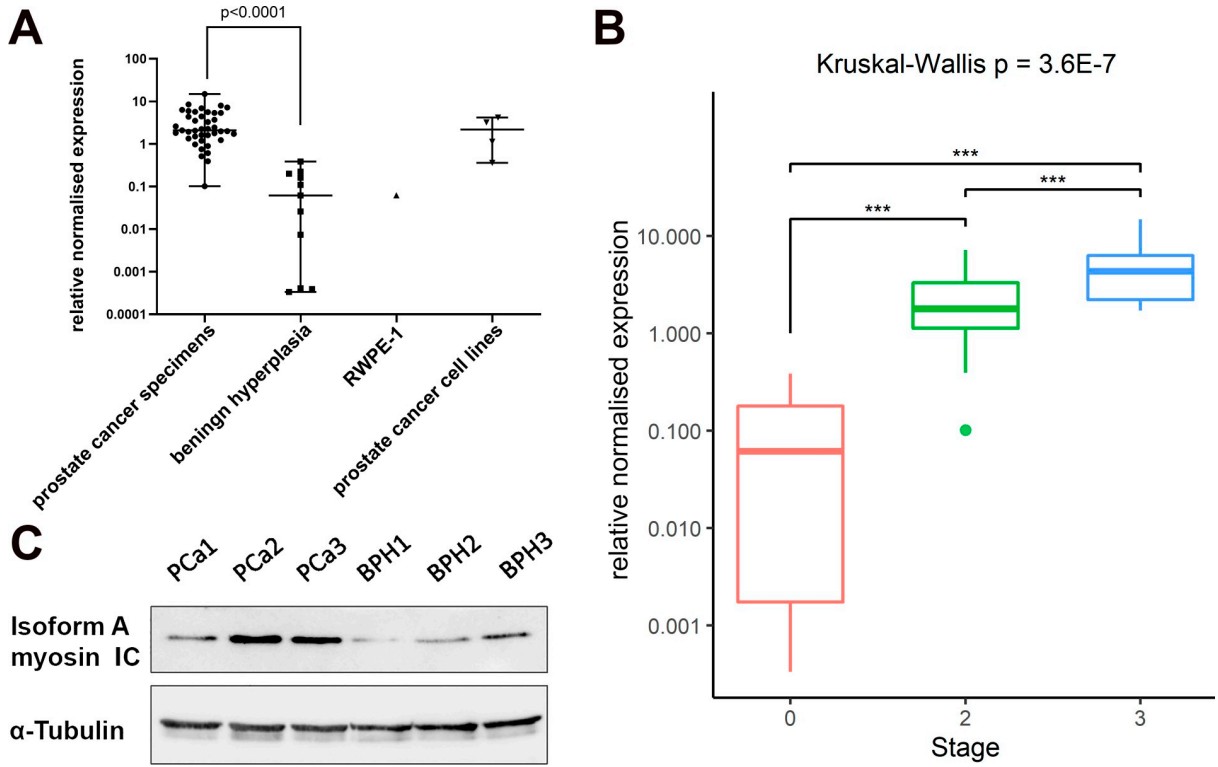

**Fig 1. Myosin 1C isoform A is overexpressed in prostate cancer specimens on mRNA and protein level.** A–relative normalized mRNA quantity in clinical BPH and PCa specimens and prostate cell lines. Horizontal lines represent range and median; all experiments were conducted in triplicate and the results normalized to YWHAZ, GAPDH, and HPRT1 genes. B—Box plot of myosin 1C isoform A levels at different disease stages. Pairwise comparison of means is indicated with p-values obtained in Conover-Iman post-hoc test with Holm correction. Data shown with median values (thick horizontal line), interquartile ranges (boxes), total ranges (whiskers), and outliers (dots). C–Immunoblotting of myosin 1C isoform A in prostate cancer samples (PCa1-PCa3) and benign prostate hyperplasia samples (BPH1-BPH3). Alpha tubulin was taken as technical control.

embedded (FFPE) prostate samples. The sample set consisted of 2 clinical cases of stage 2 (T2N0M0) prostate carcinoma, 3 cases of disease stage 3 (T3N0M0), and normal prostate tissue from 2 different individuals without cancer. Histology of the samples was verified by routine microscopy (S2 Fig). qPCR of RNA extracted from the same FFPE tissue cores as used for microscopy showed a qualitative tendency toward an increased expression of isoform A at disease stage II and a statistically significant ($p<0.05$ by the confidence interval method) increase by a factor of 135 at disease stage III relative to the normal prostate (S3 Fig). These data demonstrate that the overall pattern of myosin IC isoform A expression associated with the

**Table 1. Normalized relative myosin 1C isoform A mRNA expression in clinical samples compared to prostate cancer cell lines.**

|  | Prostate cancer clinical specimens | Benign hyperplasia specimens | RWPE-1 non-cancer prostate cell line | Prostate cancer cell lines |
|---|---|---|---|---|
| Number of values | 41 | 11 | 1 | 4 |
| Minimum | 0.102 | 0.0003 | 0.062 | 0.359 |
| 25% percentile | 1.418 | 0.0004 | 0.062 | 0.551 |
| **Median** | 2.078 | 0.0617 | 0.062 | 2.167 |
| 75% percentile | 5.361 | 0.1996 | 0.062 | 3.932 |
| Maximum | 14.89 | 0.3855 | 0.062 | 4.174 |

development of prostate cancer, which has been uncovered here on fresh clinical samples, can be followed also on archival FFPE pathology material.

## Basic phenotype of prostate cancers

The surface phenotype of prostate cancer samples was accessed using the basic 5-color antigen panel reported previously for the description of the prostate cancer and non-cancer cell lines [12] that included CD44 (marker of basal cells), CD57 (marker of luminal cells), CD24 (marker of differentiating cells), CD90 and CD133 (stromal markers). All cell lines tested and biopsy samples did not express CD133, CD57 and CD90 surface markers.

A non-malignant cell line RWPE-1 as well as prostate cancers PC-3 and DU145 demonstrated the CD44+/CD24- and CD44+/CD24+ phenotypes of progenitor cells, while prostate cancers 22Rv1 and LNCaP demonstrated CD44-/CD24- and CD44-/CD24+ phenotypes.

CD44 and CD24 were heterogeneously expressed on cells *ex vivo*, reflecting different maturation stages of the prostate epithelium. The majority of prostate cells in BPH non-cancer specimens were CD44-/CD24+ and CD44-/CD24-. The progenitor CD44+/CD24- and CD44 +/CD24+ cells–resembling a non-malignant RWPE-1 cell line were detectable as a minor fraction in BPH (Fig 2A).

Compared to non-cancer BPH tissue, prostate cancer samples demonstrated a statistically significant decrease in CD24+ cell numbers (p = 0.029, Kruskal-Wallis test). Also, there was a trend towards reduction of CD44+, although the results were not significant (Fig 2B; S4 Fig). The majority of prostate cells in cancer specimens had CD44-/CD24- phenotype with reduced CD44-/CD24+ cell fraction compared to BPH. Such phenotype corresponds to that of LNCaP and 22Rv1 cancer cell lines. None of the assessed markers were differentially expressed between stage 2 and stage 3 of prostate cancer (Conover-Iman post-hoc test with Holm correction) (Fig 2C; S4 Fig). Thus the *ex vivo* prostate cancer cells, albeit heterogeneous, predominantly corresponded to LNCaP and 22Rv1 phenotype.

## Cells with low and high isoform A expression display different surface phenotypes

To determine whether high and low myosin IC isoform A expression correlated with distinct cell surface phenotypes, we extended our panel adding seven surface molecules (CD10, CD13, CD29, CD38, CD54, CD146, and CD166) that have been associated with the tumorigenic potential of prostate cancer cells [16–22]. Five malignant and non-malignant prostate cell lines were primarily used to test the new panel. The representative dot plots are shown in Fig 3. Since cell lines exhibit more homogeneous phenotypes than patient samples, the data was analyzed as median fluorescence intensity (MFI) ratios–MFI of the stained sample divided by the MFI of the negative control (numerical data are presented in Table 2). Correlation analysis revealed that isoform A mRNA expression level negatively correlated with the presence of CD10 (Spearman R coefficient = -0.77) and CD13 (R = -0.77), and positively correlated with the presence of CD38 (R = 0.8), CD29 (R = 0.8) and CD54 (R = 0.8) Thus the expanded phenotypes are CD10+/CD13+/CD146-/CD166low/CD29low/CD54low/CD38low for LNCaP cells with the lowest *MyoICA* expression level among cell lines (normalized mRNA expression level 0.359) and CD10-/CD13-/CD146low/CD166low/CD29+/CD54+/CD38+ for PC-3 cells with the highest *MyoICA* expression level (normalized mRNA expression level 4.174).The expanded flow cytometry panel allowed the discrimination between the model cell cultures– 5 cultures into 4 groups (PC-3 and DU-145 remained similar). Thus high expression level of *MyoICA* correlated to the decreased surface levels of CD10 and CD13 and increased surface levels of CD29, CD54, and CD38 on model cell lines.

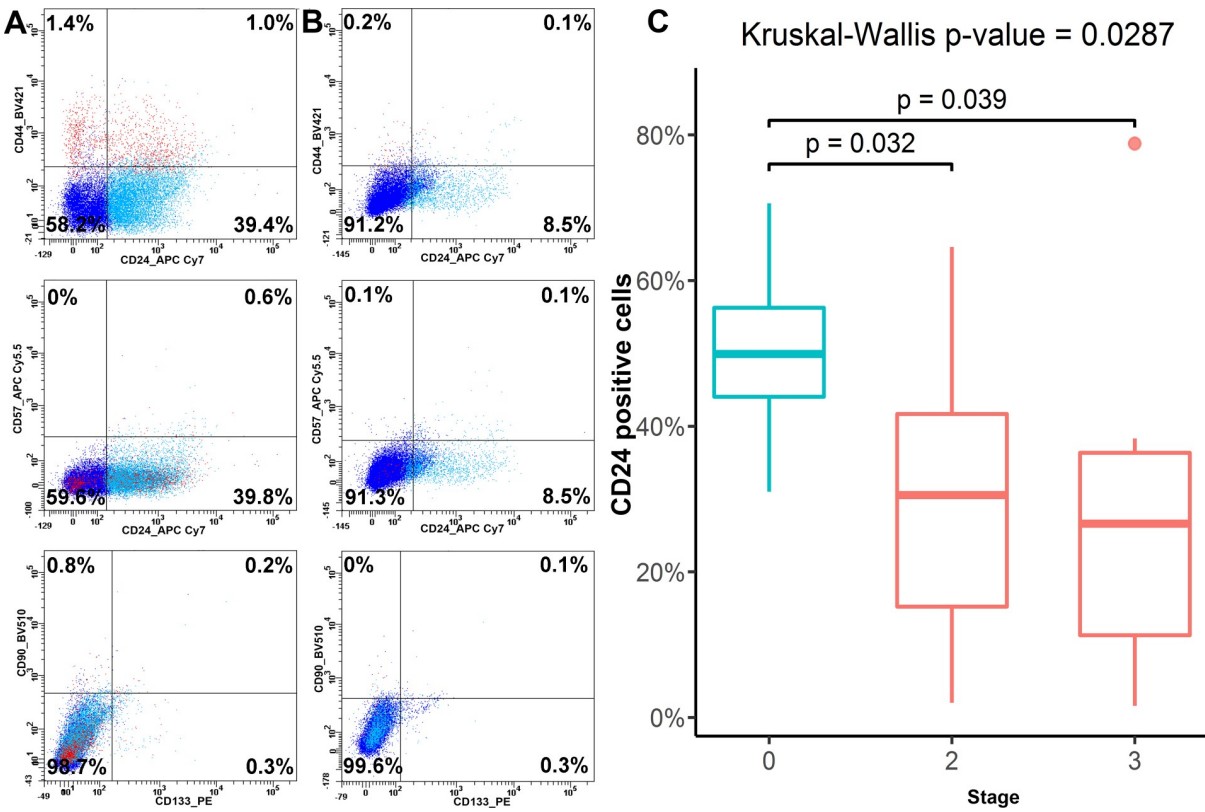

**Fig 2. Basic immunophenotype of clinical specimens.** Gates were set according to unstained controls. A—Representative immunophenotype of benign hyperplasia. The sample contains a small CD44+/CD24- subpopulation and a more significant CD24+ subpopulation with partial CD44 co-expression. The surface expression of CD133, CD57, and CD90 is low or lacking. B—Representative immunophenotype of "low-expressing" carcinoma. The surface expression of all markers: CD44, CD24, CD133, CD57, and CD90 is low or lacking. C—Percentage of CD24-postitve cells at different disease stages. Plotting as in Fig 1B.

Quadrant gates are set based on unstained controls. Rows of plots correspond to the cell lines indicated on the left. PC-3 cancer cell line demonstrates high antigen expression levels; the reference immunophenotype is CD29+high/CD54+high/CD146+/CD166+low/CD38 +high/CD10-/CD13-. 22Rv1 cancer cell line demonstrates moderate antigen expression levels; the reference immunophenotype is CD29+low/CD54+low/CD146+low/CD166+low/CD38 +low/CD10-/CD13-. RWPE-1 non-malignant prostate cell line demonstrates moderate antigen expression levels; the reference immunophenotype is CD29+low/CD54+low/CD146 +/CD166+low/CD38+low/CD10-/CD13+low.

Applying the extended staining panel to the clinical samples (15 specimens) demonstrated an overall weaker staining and very high population heterogeneity for all antigens assessed *in vitro*. *Ex vivo* prostate samples contained very few CD10+, CD13+ or CD166+ cells (maximum fraction of positive cells < 10% and median < 1.5%) corresponding to CD10-/CD13-/CD166- phenotype. The maximum fraction of CD146+ cells was 10.2% with median fraction– 4.6%, corresponding to CD146low phenotype. The fractions of CD29+ (maximum– 36.2%, median– 2.3%), CD54+ (maximum– 20.4% median– 4.9%), CD38+ (maximum– 43.5%, median– 25.8%) cells differ between samples and correspond to CD29+low/CD54+low/CD38+ phenotype (S4 Table). CD38 was the most abundantly expressed marker and similarly to CD24 demonstrated a statistically significant decrease in prostate cancer samples compared to non-cancer tissue (p<0.01, Kruskal-Wallis test). Thus, despite the intra-specimen heterogeneity,

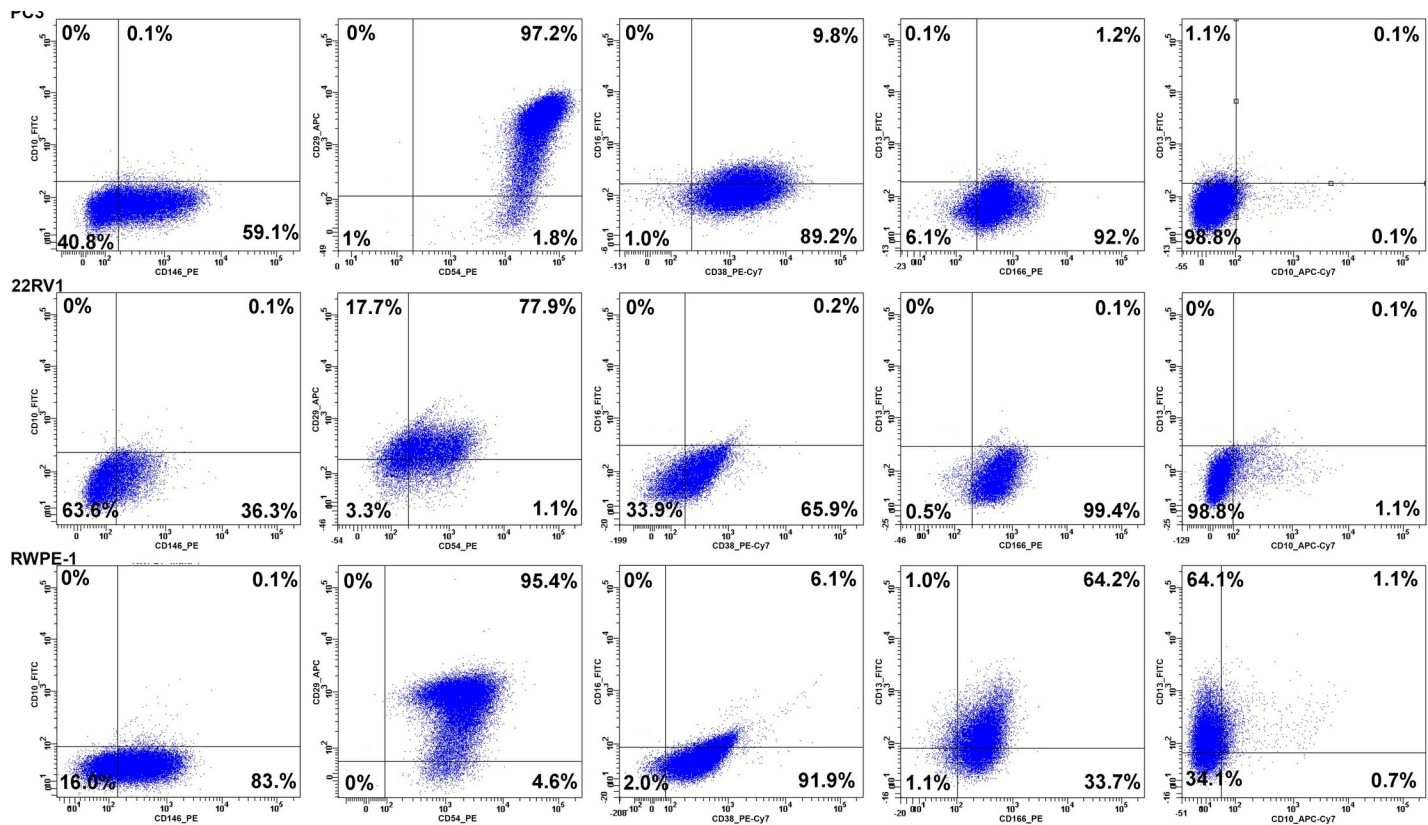

**Fig 3. Extended surface phenotypes of model prostate cell lines.**

the surface phenotype of the majority of clinical samples was similar to the prostate cancer cell lines with high isoform A expression.

Despite the overall heterogeneity, certain correlations between isoform A expression and surface phenotype were observed within sets of clinical samples pertaining to individual stages (Fig 4A). The significance of these correlations was validated by the least-squares linear and log-linear regression analysis (Fig 4B–4D). The presence of intra-stage correlations may suggest that prostate cancer cells abruptly change their surface phenotype during cancer progression, and surface markers could thus have prognostic values.

## Discussion

The need for sensitive and specific prostate cancer markers arises from the highly heterogeneous symptoms of this disease, varying from indolent to highly aggressive forms that require

**Table 2. Mean fluorescence intensity (MFI) ratio for prostate cancer cell lines.** The MFI of the stained cells was compared to the MFI of the unstained control.

| LNCaP | 22Rv1 | DU145 | PC-3 | RWPE-1 | Fluorochrome | Antibody |
|---|---|---|---|---|---|---|
| 6.1 | 9.9 | 6.6 | 6.1 | 7.7 | PE | CD166 |
| 162.4 | 1 | 1 | 1 | 1.4 | APC-H7 | CD10 |
| 6.3 | 6.1 | 768.9 | 615.6 | 46.9 | PE | CD54 |
| 12.1 | 5.6 | 33.7 | 78 | 53.2 | APC | CD29 |
| 21.8 | 5.2 | 22.7 | 34.7 | 15.2 | PE-Cy7 | CD38 |
| 1.2 | 2.1 | 3.9 | 3.2 | 6.9 | PE | CD146 |
| 2.2 | 1 | 1.6 | 1.3 | 4.7 | FITC | CD13 |

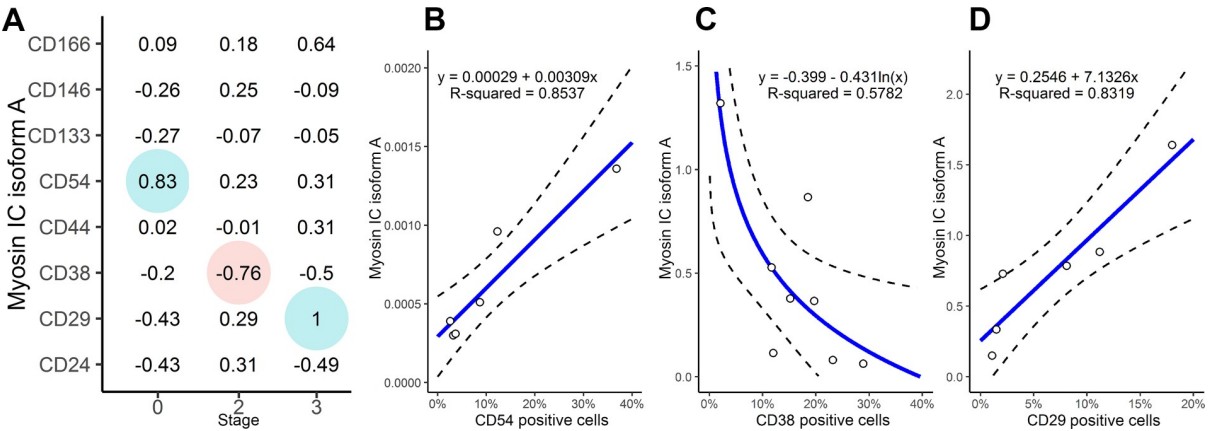

**Fig 4. Prostate cells with low and high myosin 1C isoform A expression exhibit different surface phenotypes.** A—Spearman correlation coefficients between myosin 1C isoform A expression and percentages of positive cells for indicated surface markers at different disease stages. Significant correlations ($p < 0.05$) are marked by color. B–D—Least-squares regression between myosin 1C isoform A and percentage of CD54 (B), CD38 (C) and CD29 (D) positive cells. The approximation and coefficient of determination are included. Dashed line shows the 95% confidence interval.

immediate chemotherapy and surgery. Evaluation of prostate-specific antigen (PSA) and characterization of histological material after TRUS biopsy or surgery have been set as a gold standard for PCa diagnosis, though the diagnostic value of these methods is hotly debated in the literature due to high incidence of false-positive and false-negative results [2,3,23,24]. Several novel promising mRNA markers that reduce the number of invasive diagnostic procedures and decrease the number of false results for PCa have been proposed over the years [25]. Among them, isoform A of myosin 1C has been described to be expressed specifically in progressive, metastatic prostate cancer cell lines and prostate tissue of the TRAMP mouse model with spontaneous development of orthotopic prostate cancer [9].

Myosin 1C is an unconventional single-headed non-muscle motor protein [26]. In normal epithelial MDCK cell line, it promotes the force transmission to a lateral membrane, and knockdown of myosin 1C reduces the force of cell-cell adhesion [10]. In the PC-3 prostate cancer cell line, its knockdown causes suppression of exosome secretion and an overall decrease of the cells' migratory capacity. Isoform A, specifically, is found in exosomes that contain matrix metalloproteinases MMP1 and MMP9 and is functionally involved in migration across extracellular matrix barriers [11]. Here we report that isoform A is overexpressed in clinical prostate cancer samples compared to benign hyperplasia samples both on the mRNA and protein level.

The overexpression of isoform A mRNA in cancer samples is two orders of magnitude higher than in benign hyperplasia. Thus we suggest that the use of isoform A in clinical practice will improve the sensitivity of fine needle biopsy. This is also backed by our previous data showing that isoform A allows to distinguish cancer cells in the proportion 1 to 1000 [13].

Analysis of prostate tissue biopsy is still the most reliable information for PCa diagnosis, and a multivariate model that combines several indicators increases the discrimination power but does not add considerably to the information obtained from Gleason scoring [25]. Thus, we suggest that a new candidate marker will increase the sensitivity of needle biopsy analysis and make it possible to decrease the number of core biopsies.

Validation of biomarkers suitable for non-invasive diagnostics using patients' serum and urine is an important direction in the current research on PCa [27–29]. In the light of the present findings that establish utility of myosin IC isoform A in analyzing biopsy samples and our recent results [11] that demonstrated presence of myosin IC isoform A in exosomes secreted

by PCa cells, assessing diagnostic applications of this isoform's detection in serum or urine is a logical next step in validation of this biomarker for clinical use. Expression of myosin IC isoform A, according to the data presented here, is a robust and differentiating feature of clinical samples from the different stages of the disease, and thus compares favorably with markers that are detected in small subgroups of patients (e.g., the recently characterized isoform of NCOR1, [30]). Assembly of biomarker panels with high collective diagnostic capacity is another direction of current work in the field [29]. The ability to differentiate stages of PCa progression that is demonstrated here raises the likelihood of eventually including myosin IC isoform A in a panel of biomarkers that will be capable of accurately and predictively diagnosing PCa and the likely course of the disease.

A body of work has been focused on immunophenotyping prostate cancers using immunohistochemistry as well as flow cytometry [31,32]. Despite this, no surface markers have been included in conventional panels for PCa diagnosis and staging so far. The antigens analyzed here were previously reported to be associated with distinct types of prostate cells and/or distinct stages of prostate cancer progression. CD57 is typically related to luminal epithelial cells and primary non-metastatic carcinoma [33,34], while CD44 is a basal cell and metastatic carcinoma marker [31,35]. The data indicate significance of the CD24 vs. CD44 expression pattern on various cancer cells and attribute early progenitor/stem cell properties to the CD44+/ CD24- population [36]. This pattern is particularly pronounced in breast cancer models [37–39], but some evidence has been obtained for prostate cancer models as well [40,41]. CD133 is a putative cancer stem cell marker for many cancers [42] that is transiently expressed on prostate cells and limited to the basal layer, e.g., the early progenitor cell subtype [43,44]. CD90 is a marker of prostate basal epithelial cells and stromal fibroblasts that is associated with pro-tumorigenic microenvironment [34,45]. The panel of basic antigens allowed us to discriminate the cancer and non-cancer phenotypes by CD24 expression and to show that in general, the cancer samples often demonstrate a "low-expressing" phenotype based on significantly diminished expression of CD24 (Fig 2C; S4 Fig). Although the difference between cancer and non-cancer cells was clear, there was no correlation between isoform A expression and the percentages of positive cells for any of the CD molecules from the basic panel.

Next, we sought to find whether we could attribute a specific phenotype to cells with low and high isoform A expression, using an expanded antigen panel. The expanded antigen panel included additional surface adhesion and signaling molecules: CD29 [16], CD54 [46], CD146 [18], CD166 [19], CD38 [20,21], CD10 [22,47] and CD13 [22,32], which have distinctive expression patterns and presumed tumorigenic potential in prostate malignancies. Our data on antigen expression obtained on cell lines are in accordance with Liu et al. [35]. Furthermore, we revealed that cell lines with high expression of myosin 1C isoform A have higher percentages of cells positive for specific adhesion molecules like CD29 (integrin β1), CD54 (ICAM1), CD146 (MCAM1) and CD166 (ALCAM). This suggests that cell lines with high isoform A expression may be more adhesive to the substrate, which could provide an additional mechanistic explanation for the isoform A-expressing cells' high metastatic potential.

Testing the antigen panels on clinical samples revealed a decreased antigen expression and high population heterogeneity compared to *in vitro* cell lines. All samples under study had low CD10, CD13 and CD166 percentages. CD38 demonstrated a significant (p<0.05) decrease in stage 2 and 3 prostate cancers compared with BPH (S4 Fig). The most obvious discrepancy in the phenotype of prostate cell lines and clinical samples was the expression of CD38 that positively correlated with isoform A expression in cell lines and negatively correlated with isoform A in clinical samples. While the mechanism of such shifts is unknown, the correlation pattern seen in the clinical samples may find an explanation in the light of the recent clinical data [20] where CD38 was significantly elevated in benign prostate tissue but displayed a progressively

reduced expression between the localized, metastatic, and neuroendocrine cancer subtypes, suggesting a tumor-suppressing role. The CD38 influence on NADH metabolism and related tumor signaling [21] may explain such a role, and its anti-correlation with the expression of myosin 1C isoform A is consistent with this myosin's proposed function [9,11] as a prostate cancer progression-associated molecule.

Importantly, both in prostate cell lines and the clinical dataset, we revealed a strong positive correlation between the percentage of CD54 (ICAM1)-positive cells and myosin IC isoform A expression. Among the clinical samples, this correlation was seen in benign hyperplasia, while a very strong positive correlation between the percentage of CD29 (integrin beta1)-positive cells and isoform A expression was observed in the most aggressive cases (stage 3).

Thus, the ex vivo prostate specimens contain heterogeneous cell subpopulations that correspond to different cell line phenotypes. The moderate expression levels of most surface markers and the lack of surface CD10 seen in most cancers best correspond to the phenotype of 22Rv1 model cell line. However, small subpopulations with the similar phenotype could be observed in BPH specimens. Only CD38 level (that correlated with MyoICA expression in vitro) demonstrated statistically significant differences between cancer and non-cancer specimens.

These findings may provide new criteria for differential diagnostics of prostate cancers and help to introduce FACS-based surface immunophenotyping into broad clinical practice. Moreover, they demonstrate a connection between the cell migratory potential at different clinical stages, cell adhesion patterns, and myosin expression levels, and thus have both biological relevance and prognostic value.

Taken together, our data indicate the potential utility of assessing the expression of myosin IC isoform A in diagnostics of prostate cancer, which allows distinguishing cancer from non-cancer samples even without additional sorting of stromal cells. Introduction of this marker into other gene classifier panels [48], with an additional analysis of surface antigens, will facilitate early diagnosis of prostate cancer and decrease the number of invasive diagnostic procedures.

## Supporting information

**S1 Fig. Histological patterns of the specimens with different Gleason scores.**
(TIF)

**S2 Fig. Samples stained with eosin and hematoxylin.**
(TIF)

**S3 Fig. Isoform A expression in FFPE samples, quantified by qPCR.** Normalized to total myosin IC. Error bars: standard error of the mean.
(TIF)

**S4 Fig. Surface antigen expression on prostate cancer cells *ex-vivo*.**
(TIF)

**S1 Table. Clinical data for 41 prostate cancer specimens and 11 specimens with benign prostate hyperplasia (BPH).**
(DOCX)

**S2 Table. Primer sequences for RTqPCR mRNA evaluation.**
(DOCX)

**S3 Table. List of monoclonal antibodies used for the surface staining of prostate cancer and benign hyperplasia specimens.**
(DOCX)

**S4 Table. Surface antigen expression on prostate cells from ex vivo clinical specimens and in vitro model cell lines of malignant and non-malignant origin.**
(DOCX)

**S1 Raw image.**
(JPG)

## Author Contributions

**Conceptualization:** Aleena A. Saidova, Ivan V. Maly, Wilma A. Hofmann, Ivan A. Vorobjev.

**Data curation:** Evgeniy I. Veliev.

**Formal analysis:** Galiya R. Setdikova, Daniil L. Rotin, Ivan V. Maly, Ivan A. Vorobjev.

**Investigation:** Aleena A. Saidova, Daria M. Potashnikova, Anna V. Tvorogova, Grigoriy V. Knyshinsky, Ivan V. Maly, Wilma A. Hofmann.

**Methodology:** Aleena A. Saidova, Daria M. Potashnikova, Anna V. Tvorogova, Oxana V. Paklina, Evgeniy I. Veliev, Grigoriy V. Knyshinsky, Galiya R. Setdikova, Ivan V. Maly, Wilma A. Hofmann.

**Supervision:** Ivan V. Maly, Wilma A. Hofmann, Ivan A. Vorobjev.

**Validation:** Aleena A. Saidova, Daria M. Potashnikova, Oxana V. Paklina, Daniil L. Rotin, Wilma A. Hofmann, Ivan A. Vorobjev.

**Visualization:** Anna V. Tvorogova, Galiya R. Setdikova, Ivan V. Maly, Wilma A. Hofmann.

**Writing – original draft:** Aleena A. Saidova, Daria M. Potashnikova.

**Writing – review & editing:** Aleena A. Saidova, Daria M. Potashnikova, Ivan V. Maly, Ivan A. Vorobjev.

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
