## [Decision Letter · Decision Letter 0]

3 Mar 2021

PONE-D-20-31540

Myosin 1C isoform A is a novel candidate diagnostic marker for prostate cancer

PLOS ONE

Dear Dr. Aleena Saidova,

Thank you for submitting your manuscript to PLOS ONE. After careful consideration, we feel that it has merit but does not fully meet PLOS ONE’s publication criteria as it currently stands. Therefore, we invite you to submit a revised version of the manuscript that addresses the points raised during the review process.

Reviewer #1:

Prostate Cancer is one of the leading cancers in the world and to find the new biomarkers is a very challenging task. In this paper, authors tried to find a unique and reliable biomarker for a sensitive diagnosis that will be upregulated exclusively in prostate cancer cells. In this study, Authors evaluated the expression of myosin 1C isoform A in 5 prostate cell lines, 29 prostate 30 cancer cases, and 8 benign hyperplasia tissues. Furthermore, authors studied the expression of 12 surface molecules on prostate cancer cells and performed correlation analysis with myosin 1C isoform A. After going through the paper, I noticed few things which are addressed below:

Comments1: It is not very clear in the paper, how the expression of myosin 1C isoform A was evaluated as myosin 1C isoform A has several isoforms. Do authors use any specific antibody? Please include the sequence of primers used in the study.

Comment 2: P value were not included in figure 1A

Comment 3: Since authors believe that myosin 1C isoform A could be used as a Biomarker. Does author’s tried to find the expression of myosin 1C isoform A in patent serum or urine.

Reviewer #2: 

The paper explains some interesting observations. The authors previously published a related article (PMID: 30498638). This paper reported the Myosin 1C isoform A is a novel candidate diagnostic marker for prostate cancer, I found it interesting, but authors need to explain/include some of the concerns before publications.

Concerns:

1. Add vendors' names, city, and country of origin for each instrument and software.

2. Authors need to increase the clinical sample number for a conclusive result.

3. I am wondering why the authors analyzed only cancer stage II and III samples. They should include stage I and Stage IV.

4. Authors must explain how their Myosin 1C isoform A is a better diagnostic marker than previously published biomarkers (PMID: 23308129; PMID: 33256740; PMID: 33113419; PMID: 33058520)

6. PLOS authors have the option to publish the peer review history of their article (what does this mean?). If published, this will include your full peer review and any attached files.

We look forward to receiving your revised manuscript.

Kind regards,

M. Saleem

Academic Editor

PLOS ONE

'All human material was obtained with informed consent. All procedures were performed following the ethical standards of the responsible local committee (at the Biological Department of Moscow State University) and the Helsinki Declaration of 1975 as revised in 2008.'   

(a) Please amend your current ethics statement to confirm that your named institutional review board or ethics committee specifically approved this study.

(b) Once you have amended this/these statement(s) in the Methods section of the manuscript, please add the same text to the “Ethics Statement” field of the submission form (via “Edit Submission”).

3. Please provide additional details regarding participant consent.

In the ethics statement in the Methods and online submission information, please ensure that you have specified what type you obtained (for instance, written or verbal, and if verbal, how it was documented and witnessed).

If your study included minors, state whether you obtained consent from parents or guardians.

If the need for consent was waived by the ethics committee, please include this information.

4. Please provide additional information about each of the cell lines used in this work, including any quality control testing procedures (authentication, characterisation, and mycoplasma testing). For more information, please see " ext-link-type="uri" xlink:type="simple">http://journals.plos.org/plosone/s/submission-guidelines#loc-cell-lines."

5. In the Methods section, please provide the product number and any lot numbers of the primary antibodies purchased from chemical companies  for your study.

7. Please include captions for your Supporting Information files at the end of your manuscript, and update any in-text citations to match accordingly. Please see our Supporting Information guidelines for more information: http://journals.plos.org/plosone/s/supporting-information

---

## [Author Response · Author response to Decision Letter 0]

17 Apr 2021

Dear Dr. Muhammad Saleem,

Thank you for conducting an expert peer review of our manuscript and for your favorable decision concerning our first version. We are submitting a thoroughly revised version for your consideration. We are grateful to both reviewers for their recommendations. We have carefully reviewed the comments and revised the manuscript accordingly. Our complete response and list of the changes follow. Changes to the manuscript are shown in red.

We hope that the revised version is now suitable for publication and look forward to hearing from you soon.

Kind regards, 

Aleena Saidova, PhD

Senior Research fellow

School of Biology, Moscow State University

Reviewer 1 Comment 1

It is not very clear in the paper, how the expression of myosin 1C isoform A was evaluated as myosin 1C isoform A has several isoforms. Do authors use any specific antibody? Please include the sequence of primers used in the study.

The gene for Myosin IC encodes three isoforms, termed isoform A, B, and C. Isoforms A contains a unique N-terminal peptide sequence of 35 amino acids that is encoded by an isoform specific exon. This unique exon sequence as well as the unique amino acid sequence allowed us to detect specifically myosin IC isoform A. 

The details of the myosin IC isoform A-specific primer that target the isoform A specific exon are included in Table S2. 

On protein level, myosin IC isoform A was identified using a monoclonal, isoform A-specific antibody that was raised specifically against the N-terminal isoform A-specific peptide. The antibody is described in Ihnatovych et al., 2012. The antibody has also been licensed to EMD Millipore as an isoform A specific antibody under the name: Anti-MYO1C Antibody, clone B7H8. A better description of this antibody has been added to the “method” section

Reviewer 1 Comment 2

P value were not included in figure 1A

 We have revised figure 1A and included p-value.

Reviewer 1 Comment 3

Since authors believe that myosin 1C isoform A could be used as a Biomarker. Does author’s tried to find the expression of myosin 1C isoform A in patent serum or urine.

The possibility that the reviewer raises is an important one for further development of this biomarker. While detection of myosin IC isoform A in serum or urine is outside the scope of this paper, which for the first time clearly establishes clinical relevance of this marker and focuses on its use in analyzing biopsy material, we are planning to assess its use in non-invasive diagnostics in the near future and have included the following text in the Discussion:

“Validation of biomarkers suitable for non-invasive diagnostics using patients’ serum and urine is an important direction in the current research on PCa (see, e.g., https://pubmed.ncbi.nlm.nih.gov/23308129/, https://pubmed.ncbi.nlm.nih.gov/30129068, https://pubmed.ncbi.nlm.nih.gov/33256740/). In the light of the present findings that establish utility of myosin IC isoform A in analyzing biopsy samples and our recent results (https://pubmed.ncbi.nlm.nih.gov/28814772/) that demonstrated presence of myosin IC isoform A in exosomes secreted by PCa cells, assessing diagnostic applications of this isoform’s detection in serum or urine is a logical next step in validation of this biomarker for clinical use.”

Reviewer 2 Comment 1

Add vendors' names, city, and country of origin for each instrument and software.

We have revised the Materials and Methods and added vendor details for instruments and software

Reviewer 2 Comment 2

Authors need to increase the clinical sample number for a conclusive result.

We have increased the clinical dataset on 15 samples, that allowed us to discriminate II and III clinical stages based on isoform A expression.

Reviewer 2 Comment 3

I am wondering why the authors analyzed only cancer stage II and III samples. They should include stage I and Stage IV.

Stage I is almost impossible to include since biopsies are very rarely done at this stage. Similarly, about stage IV – usually the patients are coming to the hospital at earlier stage (II or III) and biopsy cannot be postponed to follow cancer progression due to ethical reasons. Moreover, the common protocol at stage IV is chemotherapy to reduce the cancer mass that would be then classified as stage III.

Reviewer 2 Comment 4

Authors must explain how their Myosin 1C isoform A is a better diagnostic marker than previously published biomarkers (PMID: 23308129; PMID: 33256740; PMID: 33113419; PMID: 33058520)

We have added the following passage to the Discussion, which references the literature indicated by the reviewer:

“Validation of biomarkers suitable for non-invasive diagnostics using patients’ serum and urine is an important direction in the current research on PCa (see, e.g., https://pubmed.ncbi.nlm.nih.gov/23308129/, https://pubmed.ncbi.nlm.nih.gov/30129068, https://pubmed.ncbi.nlm.nih.gov/33256740/). In the light of the present findings that establish utility of myosin IC isoform A in analyzing biopsy samples and our recent results (https://pubmed.ncbi.nlm.nih.gov/28814772/) that demonstrated presence of myosin IC isoform A in exosomes secreted by PCa cells, assessing diagnostic applications of this isoform’s detection in serum or urine is a logical next step in validation of this biomarker for clinical use. Expression of myosin IC isoform A, according to the data presented here, is a robust and differentiating feature of clinical samples from the different stages of the disease, and thus compares favorably with markers that are detected in small subgroups of patients (e.g., the recently characterized isoform of NCOR1, see PMID: 33058520). Assembly of biomarker panels with high collective diagnostic capacity is another direction of current work in the field (see, e.g., https://pubmed.ncbi.nlm.nih.gov/33256740/). The ability to differentiate stages of PCa progression that is demonstrated here raises the likelihood of eventually including myosin IC isoform A in a panel of biomarkers that will be capable of accurately and predictively diagnosing PCa and the likely course of the disease.”

---

## [Decision Letter · Decision Letter 1]

7 May 2021

Myosin 1C isoform A is a novel candidate diagnostic marker for prostate cancer

PONE-D-20-31540R1

Dear Dr. Aleena Saidova,

We’re pleased to inform you that your manuscript has been judged scientifically suitable for publication and will be formally accepted for publication once it meets all outstanding technical requirements.

Kind regards,

Mohammad Saleem

Academic Editor

PLOS ONE

Additional Editor Comments (optional):

Reviewers' comments:

Reviewer's Responses to Questions

**Comments to the Author**

1. If the authors have adequately addressed your comments raised in a previous round of review and you feel that this manuscript is now acceptable for publication, you may indicate that here to bypass the “Comments to the Author” section, enter your conflict of interest statement in the “Confidential to Editor” section, and submit your "Accept" recommendation.

Reviewer #2: All comments have been addressed

2. Is the manuscript technically sound, and do the data support the conclusions?

Reviewer #2: Yes

3. Has the statistical analysis been performed appropriately and rigorously? 

Reviewer #2: Yes

4. Have the authors made all data underlying the findings in their manuscript fully available?

Reviewer #2: Yes

5. Is the manuscript presented in an intelligible fashion and written in standard English?

Reviewer #2: Yes

6. Review Comments to the Author

Reviewer #2: The revised manuscript is improved far than the previous version. I accept this modified version of the paper.

7. PLOS authors have the option to publish the peer review history of their article (what does this mean?). If published, this will include your full peer review and any attached files.

Reviewer #2: **Yes: **Hifzur R Siddique

---

## [Editor Report · Acceptance letter]

12 May 2021

PONE-D-20-31540R1 

Myosin 1C isoform A is a novel candidate diagnostic marker for prostate cancer 

Dear Dr. Saidova:

I'm pleased to inform you that your manuscript has been deemed suitable for publication in PLOS ONE. Congratulations! Your manuscript is now with our production department. 

Kind regards, 

on behalf of

Dr. Mohammad Saleem 

Academic Editor

PLOS ONE